# Quantifying neurotransmitter secretion at single-vesicle resolution using high-density complementary metal–oxide–semiconductor electrode array

Kevin A. White [1] & Brian N. Kim [1,2 ✉]

Neuronal exocytosis facilitates the propagation of information through the nervous system pertaining to bodily function, memory, and emotions. Using amperometry, the sub-millisecond dynamics of exocytosis can be monitored and the modulation of exocytosis due to drug treatment or neurodegenerative diseases can be studied. Traditional single-cell amperometry is a powerful technique for studying the molecular mechanisms of exocytosis, but it is both costly and labor-intensive to accumulate statistically significant data. To surmount these limitations, we have developed a silicon-based electrode array with 1024 on-chip electrodes that measures oxidative signal in 0.1 millisecond intervals. Using the developed device, we are able to capture the modulation of exocytosis due to Parkinson's disease treatment (L-Dopa), with statistical significance, within 30 total minutes of recording. The validation study proves our device's capability to accelerate the study of many pharmaceutical treatments for various neurodegenerative disorders that affect neurotransmitter secretion to a matter of minutes.

---

[1] Department of Electrical and Computer Engineering, University of Central Florida, Orlando, FL 32827, USA. [2] Burnett School of Biomedical Sciences, College of Medicine, University of Central Florida, Orlando, FL 32827, USA. ✉email: brian.kim@ucf.edu

Chemical transmissions at synapses are an essential mechanism in neuronal networks, because they facilitate the propagation of information throughout the nervous system for body functions, memories, and emotions. These chemical transmissions relay information between neurons through the secretion of membrane-bound neurotransmitters, which exist in high concentrations within vesicles. Neurotransmitters are released in quantal events through a fusion pore, which are created when vesicles fuse with the plasma membrane during the exocytosis process. Quantal events are fast, typically occurring in milliseconds or less, and the quantal size is small, containing less than attomoles of neurotransmitter molecules per quantum (vesicle)[1,2]. An electrochemical recording using amperometry can reveal rich details of individual quantal events, including the submillisecond dynamics of quantal secretion, quantal size (no. of molecules per vesicle), frequency, as well as the kinetics of vesicle fusion[3]. In the amperometric setup, an electric potential is held between two electrodes, a reference electrode and a working electrode. This setup allows electroactive molecules, such as dopamine, serotonin, epinephrine, and norepinephrine, to undergo either oxidation, the release of electrons from the molecule, or reduction, the acquisition of electrons to the molecule, and generates a current that is measurable.

Amperometric recordings using carbon fiber electrodes (CFEs) have been pivotal in studying the fundamental mechanisms of vesicle fusion[2,4–7] over many decades. Amperometry using CFEs have also been crucial for studying the molecular effects of neurological diseases and pharmacological treatments that modulate the exocytosis process[8–10]. The phenomenon of incomplete exocytosis events allowing recycling of vesicles for rapid reuse is called "kiss-and-run" and has been studied extensively as this secretion regulation provides insights into neuronal communications and potentially the mechanisms of neuronal plasticity[4]. The study of this phenomenon is enabled by amperometry because these events require an enhanced temporal resolution to study these millisecond scale kiss-and-run events at a single-vesicle scale[2,5,11–14]. A protein that has been studied is alpha-synuclein, which has been implicated in the pathogenesis of Parkinson's disease (PD) and has been found to reduce the frequency of exocytotic events by inhibition of vesicle priming for fusion with the cellular membrane[10]. In addition to studying the vesicle-level effects of neurodegenerative diseases, amperometry is used to study or monitor biomarkers that can reveal signs of a neurological disease. The high sensitivity and temporal resolution of amperometry has enabled in vivo animal study of monitoring the concentration of nitric oxide, which is linked to the neuronal loss[15].

Despite the significance of the studies conducted and the discoveries made using amperometry, the CFE technique is labor-intensive[16–22], time consuming[19,23–25], and can take from weeks to several months[26] to research the effects of neurological diseases or pharmacological treatments. In these studies, it is important that amperometry is performed at the single-cell level to account for heterogeneity in the characteristics of quantal release among the cell population[3,27]. Therefore, to compare the kinetics of quantal release and the size of vesicle secretion with statistical significance, typically more than 30 independent cell measurements are collected for both the experimental and control groups to develop meaningful conclusions. The prohibitive features of single-cell amperometry have limited the wide application of this technology to survey (or screen) the molecular effects of pharmacological modulations and neurodegenerative pathologies associated with the synaptic neurotransmitter secretion process.

To overcome the complications of traditional single-cell studies, scalable complementary metal–oxide–semiconductor (CMOS)-based biosensors are being developed with integrated on-chip electrode arrays[28,29]. Developed CMOS-based biosensors have demonstrated their parallel recording capabilities through the mapping of the concentration of ferricyanide across a $16 \times 16$ electrode array[30], the diffusion of $H_2O_2$ in a $32 \times 32$ electrode array[31], and the flow injection of several analytes across a $128 \times 128$ electrode array[32]. Furthermore, these CMOS-based bioelectronic systems have been developed and used for high-throughput recordings of neurotransmitter secretion from bovine chromaffin cells with a $10 \times 10$ electrode array[26,28] and from human neuroblastoma cells (SH-SY5Y) using a $32 \times 32$ electrode array[29]. Using the $10 \times 10$ electrode array, the effects of the antidepressant drugs, bupropion and citalopram, on neurotransmitter secretion have been studied[26].

In this paper, we will discuss the development of a novel silicon-based electrode array, which integrates 1024 on-chip microelectrodes (Fig. 1a, b), and is capable of single-vesicle resolution recordings. The designed silicon-based electrode array is based on CMOS technology, which enables a large-scale integration of amplifiers and other electronic circuits. An on-chip microelectrode array is integrated directly on the surface of the designed CMOS chip using additional steps of fabrication, post-CMOS fabrication. The presented post-CMOS processing has been modified from a previous work[29] to improve the noise performance of each electrochemical sensor through the use of a silicon dioxide insulation layer that defines the effective area of the electrodes throughout the array, thus permitting the developed system to record the small amperometric signals from single-vesicle secretions. With the low-noise amplifiers in the CMOS chip and the improved noise performance due to the new post-CMOS process, this work presents the first single-vesicle recordings from pheochromocytoma (PC-12) cells using monolithically integrated CMOS biosensors. The presented silicon-based bioelectronics system can capture a large pool of quantal events in the span of minutes, providing rapid results compared to the conventional amperometry technique that can take from weeks to months to amass a comparable dataset[26]. Using data collected from two experiments, the statistical analysis presented reveals the changes in the characteristics of vesicle-membrane fusion and neurotransmitter release between a control group and L-Dopa-treated group of PC-12 cells. The presented device provides a platform that can be used to closely monitor any pharmaceutical modifications to the synaptic transmission process by PD treatments, Alzheimer's disease treatments, alpha-synuclein deposits, depression therapies, and other therapies for neurodegenerative disorders, at the single-vesicle level.

## Results

**High-density silicon-based electrochemical microelectrode array for quantal analysis.** The microelectrode array is designed with each electrochemical sensor having a dedicated amplifier that permits electrochemical detection as conceptually illustrated in Fig. 1a. The CMOS device is fabricated using a standard 4-metal 2-poly 0.35 μm CMOS process and contains an array of $32 \times 32$ electrodes with an integrated amplifier array[29] (Fig. 1c). To detect the presence of neurotransmitters at an electrode, the amplifier integrates the electrons released from the electroactive molecules onto a capacitor, which results in a current-dependent voltage output, $V_{oxidation}$ (Fig. 1a). After the integration cycle, the output voltage is sampled and the charge across the capacitor is reset by a switch to initiate the next cycle. For efficient readout of the entire electrode array, there are 32 multiplexers (Fig. 1c) that handle the readout of the 32 columns of electrodes using a time-division multiplexing scheme. The multiplexers select one amplifier in the column for readout, while the other 31 amplifiers are still in their integration phase and operate in a sequentially

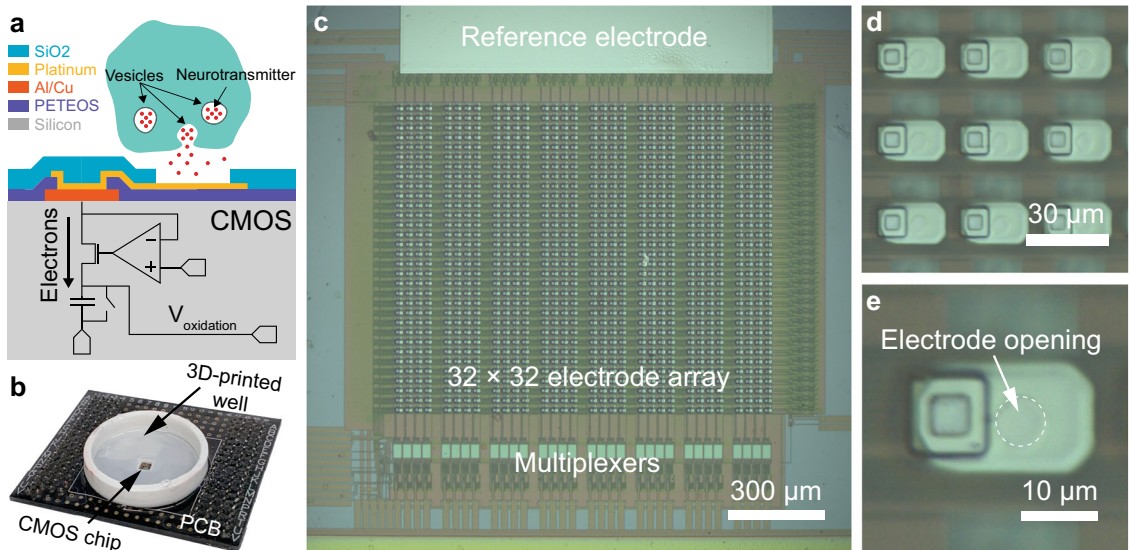

**Fig. 1 CMOS-based electrochemical detector array for high-throughput quantal analysis. a** Neurotransmitters released from cells diffuse onto an electrode and undergo a redox reaction. The oxidation releases electrons into the electrode and results in electrical current at the amplifier, which allows a buildup of electrons onto one side of a capacitor, creating a current-dependent output, $V_{oxidation}$. The output of the amplifier is readout periodically and the capacitor is reset using a switch to begin the next detection period. **b** The device is bonded onto a PCB to interface the CMOS device to external electronics. The final packaging of the device includes a well that permits the addition of electrolytic solutions to the top of the electrode array for electrochemical experiments. **c** The CMOS-based electrochemical device contains an array of 32 × 32 electrode array wherein each electrode is connected to its own dedicated amplifier and a reference electrode is integrated on-chip next to the electrode array. To readout the 1024 parallel electrochemical detections of secreted neurotransmitters, each column is connected to a dedicated multiplexer. The multiplexers stagger the readout time for the amplifiers in its dedicated column, permitting the other 31 amplifiers to operate without dead time. **d** A microphotograph of the 23 μm × 13 μm electrodes (white-colored rectangle) patterned throughout the array. **e** The oxide insulation layer is etched to provide an opening of ~5 μm above the platinum electrodes, providing an effective electrode area of 5 μm, which is comparable to the diameter of carbon fibers used for traditional single-cell electrophysiology techniques.

staggered readout pattern to minimize integration dead time and maximize their sensitivity.

CMOS fabrication foundries typically use an aluminium–copper (Al/Cu) alloy for the top metal layer, which is inadequate for neurotransmitter detection as it is highly reactive to electrolytic solutions. This highly reactive metal alloy results in significant current offsets, thus increasing the shot noise of the amplifiers. To make the electrode array suitable for neurotransmitter detection, platinum electrodes are integrated onto the chip to provide a polarizable electrode material (Fig. 1c–e).

To integrate the new platinum electrodes onto the chip's surface, a post-CMOS processing is performed using a standard photolithography process. The chip is first processed using a negative photoresist, NR9-1500PY, to create a sacrificial layer that defines where metal deposition is undesirable. After creating the sacrificial layer, 20 nm of titanium and then 200 nm of platinum are deposited using sputtering (Kurt J. Lesker Multi-Source RF and DC Sputter System, Kurt J. Lesker Company, Jefferson Hills, PA). The sacrificial layer of photoresist is then removed by rinsing the chip with acetone. Once all of the photoresist is removed, the platinum electrodes are patterned as 23 × 13 μm rectangles as shown in Fig. 1d. A 300 nm passivation layer of silicon dioxide is then deposited onto the chip using a plasma-enhanced chemical vapor deposition machine (STS 310PC, STS Ltd., Wales, UK). To reopen the electrodes for neurotransmitter detection the chip undergoes another photolithography process using the negative photoresist NR7-1500P, resulting in a protective layer that only permits the etching of the oxide above the electrodes. After the protective layer is processed, the device is etched using a reactive ion etcher with inductively coupled plasma (Unaxis SLR, Unaxis Materials Inc., Santa Clara, CA). The etching process produces circular electrode openings through the

oxide passivation layer that are ~5 μm in diameter as shown in Fig. 1e. To remove the remaining NR7 photoresist and clean the surface of the array, the device is put into a piranha solution (3:1 mixture of sulfuric acid:hydrogen peroxide). The resulting effective area of the electrodes throughout the array is comparable to the gold standard CFEs with diameters of ~5 μm[27,33–36].

After all post-CMOS processing is completed, the chip is packaged to integrate a reservoir structure that holds electrolytic solutions on the surface of the array and prevent damage to the wire bonds and interfacing electronics. The chip is first bonded to a custom-designed PCB to interface with a custom-designed multifunctional data acquisition system. Then, a 3D-printed ABS plastic well that is coated in polydimethylsiloxane (PDMS) is bonded to the surface of the electrode array using additional PDMS. The fully packaged device is shown in Fig. 1b. After the device is packaged, the electrode array is coated with poly-D-lysine (PDL) to enhance the adhesion of cells to the electrode array during experiments as described in the "Methods" section.

The amplifiers throughout the array have an average noise level of ~0.42 $pA_{RMS}$ when the device is operating without electrolytic solution at a 10 kHz sampling rate and a bandwidth of ~4.4 kHz[29]. In comparison, a recent CMOS device has been developed for high-throughput recording of electrochemical current with an array of amplifiers that exhibit a noise level of 1.1 $pA_{RMS}$ at a sampling rate of ~10 kHz with a bandwidth of ~2 kHz[37]. An electrode that is in contact with electrolytic solution suffers from an elevated noise level due to the introduced double-layer capacitance. Our previous design had an electrode area of 15 μm × 15 μm, and when electrolytic solution was placed onto the array for electrochemical experiments the noise level was 6.16 $pA_{RMS}$[29], which is higher than the average amplitude of PC-12 quantal events. The new post-CMOS processing presented in this

paper creates an electrode with a diameter of ~5 μm and produces an average noise of ~0.9 pA$_{RMS}$ under the same conditions. In comparison, a study using a 5 μm diameter CFE with the Axopatch 200B amplifier, a high-quality electrophysiology amplifier, had a noise level of ~1.4 pA$_{RMS}$ with a 10 kHz bandwidth[34]. Our electrochemical sensor array exhibits a comparable noise level to a high-quality electrophysiology setup, therefore enabling high-throughput recordings of quantal events from dopaminergic cells, such as PC-12 cells.

Based on the noise level of the presented device (~0.9 pA$_{RMS}$) and the typical half-width of amperometric spikes recorded from PC-12 cells (~5 ms), we can estimate the smallest quantal size that can be detected using this system. Because the integrated noise level for 5-ms events is about 0.14 pA$_{RMS}$, the quantal size equivalent noise level for 5-ms signal is ~2000 molecules.

**On-chip recordings at single-vesicle resolution**. Cells are loaded onto the surface of the electrode array as described in the "Methods" section. Loaded cells are settled for 5 min to the surface of the array, then unattached cells are washed away. An example of randomly distributed cells over the electrode array is shown in Fig. 2a. The cells are present in single-cell form with diameters of 5–10 μm, as well as in clumps (Fig. 2b, c), throughout the array. Quantal events from the attached cells are recorded in parallel (Fig. 3a) with the electrodes throughout the array held at a potential of ~800 mV with respect to the reference electrode and the amplifiers operating at a 10 kHz sampling rate.

After a set of recordings are acquired with the device, the recorded exocytotic spikes (Fig. 3b) are analyzed using a quantal analysis program, Quanta Analysis v8.20[3]. The analyzed characteristics of an exocytotic spike include the maximum amplitude of the signal ($I_{max}$), the duration of the signal at half of its maximum height ($t_{1/2}$), and the quantal size ($Q$) as illustrated in Fig. 3c. After 14 min of recording, 477 quantal events are detected from the control group (Fig. 4). In the control group, the average amplitude of the quantal events is 1.30 picoampere (pA), the average half-width of these events is 7.33 ms, and the average quantal size is 14.3 fC.

**Rapid characterization of PD therapy on quantal release in minutes**. L-Dopa is used clinically for PD because it is known to increase the dopamine concentration in the central nervous system[9]. To validate the single-vesicle recording capability of the presented device, amperometric spikes for both untreated and L-Dopa-treated cells are measured and analyzed at the single-vesicle level. For untreated cells, the chip measured from a total number of 16 cells and 477 quantal spikes in a 14-min recording. For the L-Dopa-treated cells, a total number of 63 cells and 3534 vesicle secretions are measured, also in a 14-min recording period. L-Dopa-treated cells have a noticeably higher number of quantal events compared to the untreated cells and the amplitude of each

spike tends to be larger (Fig. 4a, b). The quantitative analysis on both treated and untreated cell recordings reveals significantly altered quantal release characteristics (Fig. 4c–f) from these short recordings. Mainly, the amplitude and the quantal size are increased with the L-Dopa treatment as anticipated because of L-Dopa's role as a dopamine precursor. The average spike amplitude for treated cells is 1.75 pA, as opposed to 1.30 pA of untreated cells (Fig. 4c). The half-width increased slightly after the L-Dopa treatment from 7.33 to 9.15 ms (Fig. 4d). The L-Dopa treatment increased the average quantal size by 86%, with the average quantal sizes being 26.6 and 14.3 fC for treated and untreated cells, respectively (Fig. 4e). The number of electroactive molecules contained within each vesicle is calculated as described in the "Methods" section. The average number of molecules measured per vesicle release for treated and untreated cells, respectively, are 83,100 and 44,600 molecules (Fig. 4f).

All recorded quantal events from both cell groups are also compared in histograms (Fig. 4g–j). Secretory cells regulate the vesicles in diameters (φ), and we expect to see a well-fitted Gaussian distribution of vesicle's diameter[38]. Quantal size is proportional to the volume of vesicles (φ³) and plotting of (quantal size)$^{1/3}$ should produce a Gaussian distribution. As expected, the quantal size plots in attomole$^{1/3}$ show good agreements with Gaussian distributions for both treated and untreated cells as seen in Fig. 4j. The average quantal size in attomole$^{1/3}$ for L-Dopa-treated cells is $0.50 \pm 0.11$ attomole$^{1/3}$ (mean ± SD) and is $0.40 \pm 0.08$ attomole$^{1/3}$ (mean ± SD) for the control group.

## Discussion

Using the presented CMOS integrated electrochemical sensors, we are able to perform hundreds of amperometric recordings in minutes as opposed to weeks using the conventional CFE approach[26]. Significant cost reduction in instrumentation for quantal release analysis will allow the rapid evaluation of existing and emerging treatments for their molecular effects on exocytosis.

Using the integrated silicon device, we were able to characterize the effects of L-Dopa treatment at the single-vesicle level, and our results agreed with previous studies[9,33,39–42]. After treating PC-12 cells with L-Dopa, we observed a 35% increase in $I_{max}$, which is similar to the previous reports[39,41]. Modulation of $t_{1/2}$ as a result of L-Dopa treatment has increased the duration of quantal events by 47% in MN9D cells[1] and by 10%[39], 38%[41], and as large as 88%[42] in PC-12 cells. Our results showed a similar increase in $t_{1/2}$ of 25% after L-Dopa treatment. Across all studies, L-Dopa treatment increases the quantal size within each vesicle; treated bovine chromaffin cells exhibited a 70% increase[43]; MN9D cells exhibited an increase of 40%[1]; and PC-12 cells exhibited an increase of 20[39], 26[42], and 63%[41]; and our measurement showed an 86% increase in quantal size.

The presented data, specifically the validated changes in $I_{max}$, $t_{1/2}$, and $Q$ quantal characteristics after L-Dopa treatment,

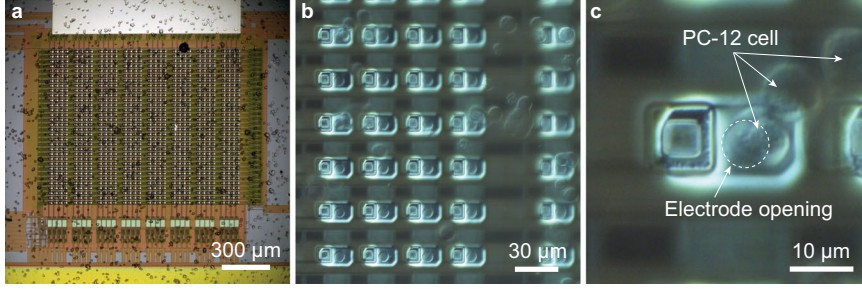

**Fig. 2 On-chip recording of PC-12 cells. a** The settled cells on the surface of the device. **b** Cells are seen in single-cell form, as well as clumps of cells, throughout the electrode array. **c** Cells have settled atop the electrode openings, permitting electrochemical monitoring of quantal secretion.

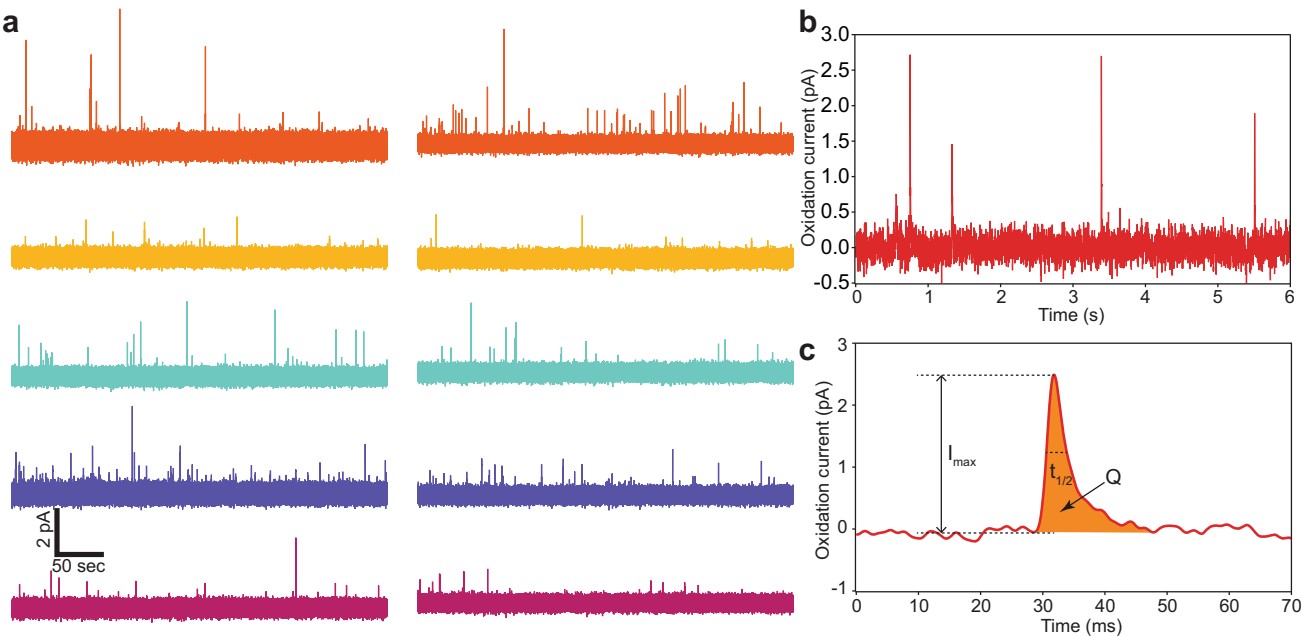

**Fig. 3 Parallel neurotransmitter secretion recordings. a** Representative amperometric recordings from a single recording session. **b** A section of a recording showing the fast amperometric spikes related to vesicle release events. **c** Each exocytotic event is analyzed for key quantal release characteristics (maximum peak of the spike $I_{max}$, the half-width duration of the spike $t_{1/2}$, and the quantal size $Q$).

support the application of the presented system for studies that require single-vesicle resolution or high-throughput, and high-temporal resolution, recordings of any electroactive molecules that undergo a redox reaction below the potential that causes electrolysis of water (1.23 V). The presented work, in combination with microscopic imaging, offers single-cell monitoring capability. The flatness of the chip's surface offers an excellent environment for neuronal cells, or model cells such as PC-12, to create neural networks. Different surface structures may be ideal depending on the application and truly isolated single-cell recordings can be accomplished with minor modifications to the design, such as using a thick insulation layer to only permit single cells to each electrode. We have demonstrated this technique in our previous work using SU-8[29]. However, by incorporating this insulation layer onto a device, the array is not ideal for certain applications including the study of neural networks; the large topology of the insulation may introduce challenges when attempting to culture a neural network on the surface of the array.

The presented work enables single-vesicle level amperometric recordings from PC-12 on the CMOS-based biosensor by restricting the size of electrodes, thus reducing the noise level such that it closely matches that of the high-quality CFE electrophysiology setup. This work presents a high-throughput analysis platform for electrochemical detection of quantal secretion with single-vesicle resolution. The high-throughput simultaneous measurements of quantal release from a large number of cells are anticipated to enable rapid studies of neurodegenerative disorders and the effects of pharmacological therapies that target neurotransmitter release.

## Methods

**Recording and stimulation solutions**. The recording bath solution consists of 150 mM of NaCl, 5 mM of KCl, 2 mM of CaCl₂, 1.2 mM of MgCl₂, 10 mM of HEPES, and 11 mM of glucose. Additional glucose was added to adjust the osmolality of the recording solution to 320 mmol/kg and the pH was adjusted to 7.3 using NaOH. The high K⁺ stimulation solution consists of 55 mM of NaCl, 100 mM of KCl, 5 mM of CaCl₂, 2 mM of MgCl₂, 10 mM of HEPES, and 10 mM of glucose. The stimulation solution's osmolality was adjusted to 320 mmol/kg by adding glucose to match the osmolality of the recording bath solution and the pH was adjusted to 7.3

using NaOH. Both solutions were prepared using water from a purification system and then further filtered after preparation to ensure their sterility.

**PC-12 cell culture**. The PC-12 cells were obtained from the American Type Culture Collection (ATCC) (Manassas, VA) and were cultured using T-75 flasks in an incubator at 37 °C with 5% CO₂. The culture media consists of RPMI-1640 (Gibco) media that was supplemented with 10% heat-inactivated horse serum (Gibco), 5% fetal bovine serum (ATCC), and 1% penicillin–streptomycin (Gibco). The media were changed every 2–3 days and the cells were subcultured when the cell density was ~4 × 10⁶ cells/mL. The cells used for the control group and the L-Dopa-treated group are first split from one flask into two separate flasks the day of, or before the on-chip measurement.

**Experimental procedure**. Cells were first detached from their flasks by a washing process using culture media and the cell concentration was adjusted to ~2 × 10⁶ cells. The detached cells were first spun down to a pellet in a centrifuge at 200 × g for 5 min. The supernatant was discarded, 5 mL of the recording solution was added, and the cells are resuspended using a 20 mL syringe with a 22-gauge needle. The cells were once again spun down to a pellet at 200 × g for 5 min, the supernatant was discarded, and 1 mL of new recording solution was added. The approximate 2 × 10⁶ cells were resuspended once more and then 20 μL of this cell solution was loaded directly onto the electrode array. Once loaded onto the device, the cells were given 5 min to settle to the surface of the array. During this settling period, a prepared Ag|AgCl wire reference electrode was placed into the well. Any unattached cells were removed from the device by washing the array with more recording solution using a perfusion system. To treat the electrode array with PDL (Gibco), the device was incubated with 0.1-mg/mL PDL solution on top of the array for 1 h. The device was then rinsed thoroughly using deionized water and dried for at least 1 h before the array's first electrochemical experiment.

A brief recording of the quantal release from nonstimulated cells resting on the electrode array was performed before the high K⁺ stimulation solution was added to the electrode array. The stimulation solution was then added using a perfusion system, which was operated for about 1 min to saturate the cells with the exocytosis-inducing stimulant. The recording lasted for 14 min after the first stimulation. After the cells have rested for at least 5 min following the first stimulation, the perfusion system was used again to add more of the high K⁺ stimulant to restimulate the cells halfway through the recording period. When performing experiments with L-Dopa-treated cells, the cells are exposed to 100 μM of L-Dopa for 1 h prior to executing the cell loading process. To determine the viability of the cells during the recordings of neurotransmitter secretion, trypan blue was added to the electrode array after all recordings are complete. The solution on the array was first exchanged with fresh recording bath solution and then drained until 1 mL of solution remains. Then, 1 mL of trypan blue was added to the well to stain any dead cells atop the array. After 5 min of cell staining, the trypan blue was removed by perfusion and exchanged with new recording solution and the array was inspected using a microscope.

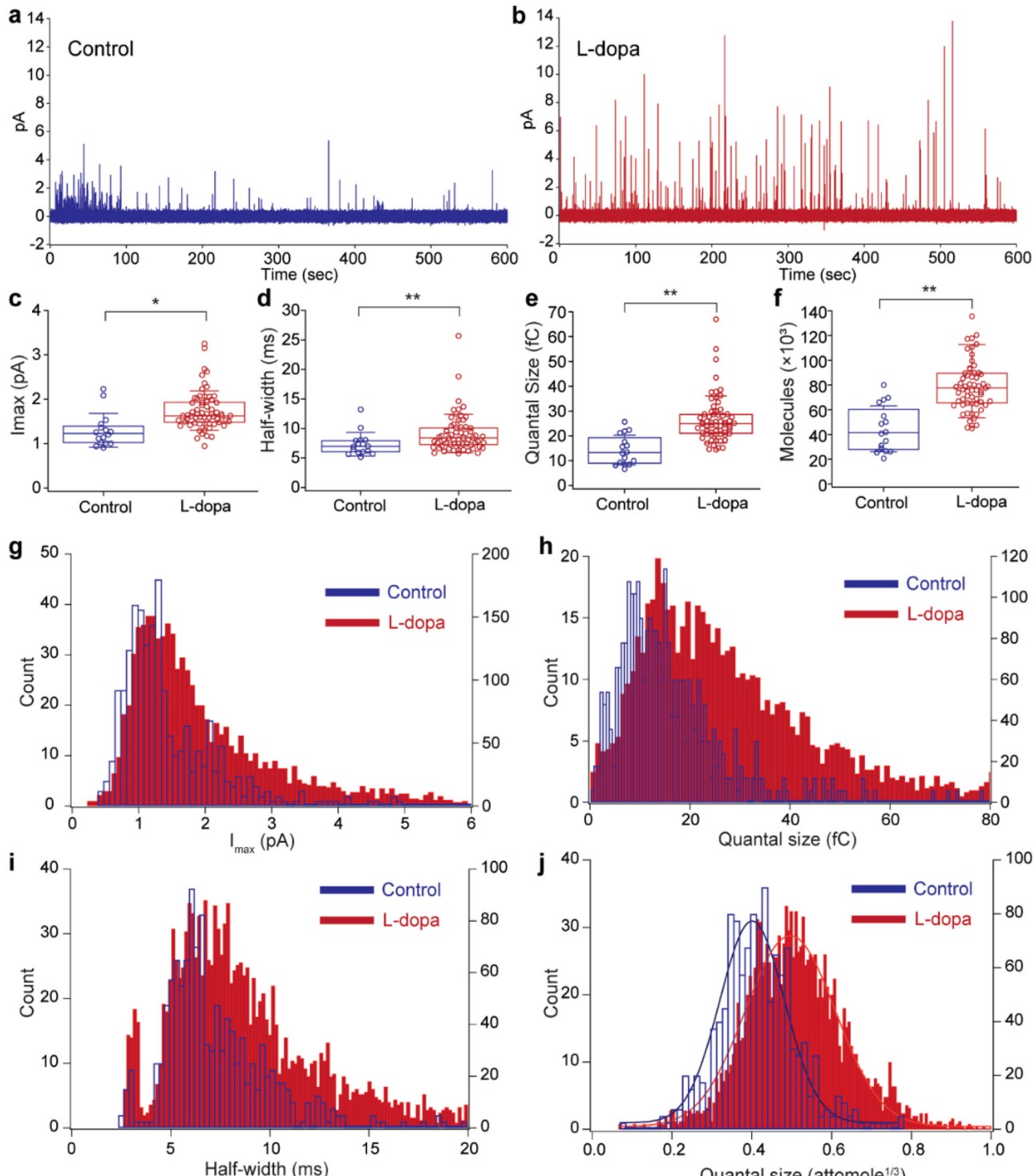

**Fig. 4 Vesicle secretion of neurotransmitter is modulated by L-Dopa treatment. a** A representative recording from the control group. **b** A representative recording after L-Dopa treatment that exemplifies an alteration in neurotransmitter secretion characteristics. **c–f** After L-Dopa treatment, the quantal release characteristics of the PC-12 cells are significantly altered ($n = 16$ for control, $n = 63$ for L-Dopa, whiskers are drawn from the quartiles to the extreme values, the box is generated from the first and third quartile as well as median). **c** The average exocytotic spike amplitude increased from 1.3 to 1.75 pA after treatment. **d** The average half-width increased after treatment from 7.33 to 9.15 ms. **e** The average quantal size increased from 14.3 to 26.6 fC after L-Dopa treatment. **f** The change in quantal size correlates to an average molecule quantity increase from 44,600 to 83,100 for treated cells. **g–j** Histogram showing characteristics from all detected spikes. (two-tailed Student's $t$ test: $*p < 0.01$, $**p < 0.001$).

**Statistical analysis**. All of the cell recordings are analyzed in Igor Pro 8 using Quanta Analysis version v8.20[3]. All recordings are filtered with a 1 Hz high pass filter, to remove low-frequency fluctuations resulted from perfusion steps, and a low pass binomial smoothing set to 100 Hz. Signals are detected as quantal events when the signal's first-order time derivate is five times greater than the standard deviation of the background noise. All individual exocytotic spikes are analyzed for their quantal size, amplitude, half-width, and number of molecules. The quantal size was calculated based on the integration of the electrochemical current induced during an exocytotic spike in picocoulomb. The amplitude is determined from the baseline to the peak of each spike in pA. The number of molecules released per

vesicle can be calculated by dividing the quantal size by 2q (q is the electron charge). Exocytotic events from individual cells are heterogeneous and over-representation is possible from cells that are highly active[40]. In order to prevent the overrepresentation of highly active cells, the median value of all the spikes from a single cell is used as n (each sample) in the statistical analysis, rather than using individual spikes as n. This method is consistent with other amperometric studies[28,29,44,45].

**Reporting summary**. Further information on research design is available in the Nature Research Reporting Summary linked to this article.

## Data availability

The data that support this work are available from the corresponding author upon reasonable request. All relevant data are publicly available (GitHub: https://github.com/bioelectronicsUCF/NatComm2020).

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

## Acknowledgements

This work was supported by the National Science Foundation under grant #1745364.

## Author contributions

Both authors conceived the research plan, designed experiments, performed research, analyzed data, wrote the initial draft, and revised the manuscript. B.N.K. supervised the project and acquired funding.

## Competing interests

The authors declare no competing interests.
