## [Peer Review File · Nature Communications]

REVIEWER COMMENTS

Reviewer #1 (Remarks to the Author):

This paper shows for the first time the use of a recently developed (Ref 22 White et al. 2018, IEEE Trans. Biomed. Circ...) 1024-channel CMOS based amperometric detector chip to analyse the modulation of quantal size of dopaminergic by a therapeutic drug (L-DOPA).

This is an important milestone and could lead to novel screening technologies for neuropharmacology. The paper would have benefited from at least one other instance (i.e. a presynaptic modulator) but is a well-conducted, seminal study that deserves presentation to a general readership.

The paper is unfortunately not well written and needs extensive editing.

Examples:

"Chemical transmission(s) is (are)", of course, not "an essential component of neurons".

Per quanta -> per quantum (sing.)

"...uses a microelectrode to measure the electrochemical current generated by the electroactive molecules" -> not a good explanation, reference to oxidation and electrode holding potential missing.

"Amperometrics recordings ...have been Imperative" -> important, key, critical, pivotal, yes, but imperative is directed to the future, so not appropriate.

"Disease/treatment modulation over many decades" for both this and the former statement, references are missing. And what is a disease/treatment modulation?

Overall, the introduction/motivation could and should be much more concise.

When discussing imitations of the method, the fact that amperometry is generally limited to transmitters that can be oxidized at voltages below the redox potential of water should not be withheld.

Reviewer #2 (Remarks to the Author):

The authors present a silicon-based electrode array for single-cell amperometry to assess the modulation of exocytosis due to drug treatment. This system can assist on the development of effective pharmaceutical treatments for various neurodegenerative disorders that affect neurotransmitter secretion. Therefore the research is timely and has a great significance.

Nonetheless, it is not clear the novelty of the work. Single cell amperometry and the use of carbon fibre electrodes, has been already widely reported. As such, the novelty lies in the development of the device and the method for rapid data interpretation. While these features are important in advancing research in the field, the level of innovation may not be at the level of publication in a journal like Nature Communications. It is also not clear what is the key novelty compared to other CMOS-based bioelectronic systems for single-cell amperometry already reported in the literature. The authors admit that their device is not capable of true single-cell monitoring (Fig. 2c) and report the issue of variability of electrode coverage by the settled cells.

While they proposed several approaches to overcome these issues, they should provide additional evidence by testing at least some of these strategies to demonstrate their effectiveness and stress the relevance of the work presented. As it stands the work looks uncomplete.

Overall the manuscript is well written, but the presentation could be improved by for example removing several repetitions along the manuscript that affect the flow. For example it is mentioned several times that L-Dopa is a popular Parkinson's disease therapy. Moreover the flow is broken by several short sentences not clearly connected to each other.

Reviewer #3 (Remarks to the Author):

In this manuscript the authors present on-chip measurements of neurotransmitter release using an amperometry CMOS chip. High throughput measurements from PC12 cells are presented showing the expected increase of quantal size after L-Dopa treatment. The described technology is based on some previous work using chromaffin cells and SH-SY5Y cells. The method is very elegant and efficient and the recordings are of very high quality. The results are convincing but unfortunately, the authors do not point out clearly where the novelty lies.

The central claim is that the technology with a 1024 on-chip electrodes performs amperometric recording with 10 kHz sampling rate and that the effect of L-Dopa treatment could be demonstrated in a 30 minute recording with very high statistical significance ($p < 0.001$ according to Fig. 4). What aspects are advanced compared to the previous work?

The authors should describe more specifically where the specific advance lies. Is the CMOS chip the same as used in their previous work with SH-SY5Y cells (White et al 2018) or are there any modifications? Has the density of electrodes been increased? Is the 5 μm electrode diameter changed from the previous work? What is the background noise? Is it lower than with carbon fibers or previous amperometry chip versions?

A potential novelty is that the measurements were performed using PC12 cells, which have much smaller quantal size than chromaffin cells. Is this the first report demonstrating on-chip amperometry recordings using PC12 cells? Is there a new record to detect small quantal size (the authors state 187 molecules). What is the "resolution", i.e. what is the error bar for individual quantal size determination ($187 \pm ???$ Molecules). It should be described clearly what the technology can achieve beyond high throughput.

Minor points:

The introduction is too long describing many results obtained with conventional amperometry, which is not highly relevant for this manuscript. For instance most of what is on page 3 could be deleted.

The paragraph preceding subheading "Identifying effect of Parkinson's disease therapy to quantal release in minutes" duplicates some of what is in the following section. It may be more logical to remove in that paragraph all text related to L-Dopa and focus on untreated cell recordings and issues of sensitivity and resolution of amperometric spike parameters.

Response to referees

Response to Reviewer #1

Comment 1

- (a) This paper shows for the first time the use of a recently developed (Ref 22 White et al. 2018, IEEE Trans. Biomed. Circ...) 1024-channel CMOS based amperometric detector chip to analyze the modulation of quantal size of dopaminergic by a therapeutic drug (L-DOPA). This is an important milestone and could lead to novel screening technologies for neuropharmacology. The paper would have benefited from at least one other instance (i.e. a presynaptic modulator) but is a well-conducted, seminal study that deserves presentation to a general readership.
- (b) We appreciate the positive response.

Comment 2

- (a) The paper is unfortunately not well written and needs extensive editing.
Examples:
- “Chemical transmission(s) is (are)”, of course, not “an essential component of neurons”.
 - Per quanta → per quantum (sing.)
 - “...uses a microelectrode to measure the electrochemical current generated by the electroactive molecules” → not a good explanation, reference to oxidation and electrode holding potential missing.
 - “Amperometric recordings ...have been Imperative” → important, key, critical, pivotal, yes, but imperative is directed to the future, so not appropriate.
 - “Disease/treatment modulation over many decades” for both this and the former statement, references are missing. And what is a disease/treatment modulation?
- (b) We agree with the comment. In response, we have made extensive revision to the writings throughout the manuscript. We have corrected all the textual issues. Also, the concept of amperometry is now clearly explained. We have also clarified any sentences relating to “modulation.”
- (c) The list of changes is below:

On page 2:

Chemical transmissions at synapses are an essential mechanism in neuronal networks, because it facilitates the propagation of information throughout the nervous system for body functions, memories, and emotions.

On page 2:

Quantal events are fast, typically occurring in milliseconds or less, and the quantal size is small, containing less than attomoles of neurotransmitter molecules per quantum (vesicle)^{1,2}.

On page 10:

→Removed sentence: “Therefore, despite its usefulness in understanding the true role of many treatments on the quanta level, the cost and time that must be invested are prohibitive with the conventional CFE approach.”

On page 2:

To measure the secretion of electroactive neurotransmitters, such as dopamine, serotonin, epinephrine, and norepinephrine, the amperometric setup consists of a reference electrode and a working electrode. An electric potential is held between the two electrodes, thus allowing electroactive molecules to undergo either oxidation, the release of electrons from the molecule, or reduction, the acquisition of electrons to the molecule, and generate a current that is measurable.

On page 2:

Amperometric recordings using carbon fiber electrodes (CFEs) have been pivotal in studying the fundamental mechanisms of vesicle fusion^{2,4-7}, as well as the molecular effects of neurological diseases and pharmacological treatments that modulate the exocytosis process⁸⁻¹⁰, over many decades.

On page 3:

Despite the significance of the studies conducted and the discoveries made using amperometry, the technique is extremely labor-intensive and can take months to years to research the effects of neurological diseases or pharmacological treatments.

Comment 3

- (a) Overall, the introduction/motivation could and should be much more concise.
- (b) We agree with the comment. In response, we have made extensive revisions in the Introduction section to make it more concise. The new introduction section contains the key points in the following bulletin.
- 1st paragraph: Importance of chemical transmission and the tools to measure it.
 - 2nd paragraph: Impact of amperometric recordings using CFE.
 - 3rd paragraph: The challenges associated with CFE.
 - 4th paragraph: Emerging CMOS biosensors to overcome the obstacle.
 - 5th paragraph: Brief outline of the presented work.
- (c) The list of major changes is below:

On page 2:

→Removed sentences: “In secretory cells, including neurons, neurotransmitters are membrane-bound at high concentration in secretory vesicles.”

“Neuronal excitation causes vesicle fusion with the plasma membrane, also known as exocytosis, in which the membrane-bound neurotransmitters are released in quantal events through a fusion pore, a nanoscopic channel that connects the vesicular lumen to the cell’s exterior.”

→New sentences: These chemical transmissions relay information between neurons through the secretion of membrane-bound neurotransmitters, which exist in high concentrations within vesicles.

Neurotransmitters are released in quantal events through a fusion pore, which are created when vesicles fuse with the plasma membrane during the exocytosis process.

On page 2:

→Removed sentences: “The amperometry technique is an electrochemical method which uses a microelectrode to measure the electrochemical current generated by the electroactive molecules, including dopamine, serotonin, epinephrine, and norepinephrine at the electrode’s surface⁴.”

“Amperometry measures the secretion event by directly oxidizing the neurotransmitter molecules, which causes a measurable electric current into the electrode.”

→New sentences: To measure the secretion of electroactive neurotransmitters, such as dopamine, serotonin, epinephrine, and norepinephrine, the amperometric setup consists of a reference electrode and a working electrode. An electric potential is held between the two electrodes, thus allowing electroactive molecules to undergo either oxidation, the release of electrons from the molecule, or reduction, the acquisition of electrons to the molecule, and generate a current that is measurable.

On page 2:

→Removed sentences: “Conventionally, amperometric recordings are performed by placing a carbon fiber electrode (CFE) at proximity to a cell, such as a dopaminergic neuron, chromaffin cell, or pheochromocytoma (PC-12) cell.”

On page 2:

→Removed sentences: “Using traditional amperometry techniques researchers have discovered that extracellular calcium ... are secreted during exocytosis events and a lack of the C-terminus leads to prolonged secretion events¹².”

“Amperometry is also used to study the effects of various treatments that modulate exocytosis characteristics, ... reduces the quantal size of the exocytotic events while curcumin accelerates the exocytotic process¹⁷.”

“To monitor the concentration of neurotransmitters, such as acetylcholine, in vivo to enhance therapies for neurological diseases, surface modified electrodes are being developed to enable amperometry monitoring of biomolecules that are not inherently electroactive¹⁹.”

Comment 4

- (a) When discussing limitations of the method, the fact that amperometry is generally limited to transmitters that can be oxidized at voltages below the redox potential of water should not be withheld.
- (b) We agree. In response, we will add information related to the electrolysis of water.
- (c) We added new text to clarify the limitation of amperometry.

On page 10:

The presented data supports the application of the presented system for studies that require single-vesicle resolution or high-throughput, and high-temporal resolution, recordings of any electroactive molecules that undergo a redox reaction below the potential that causes electrolysis of water (1.23V).

Response to Reviewer #2

Comment 1

- (a) The authors present a silicon-based electrode array for single-cell amperometry to assess the modulation of exocytosis due to drug treatment. This system can assist on the development of effective pharmaceutical treatments for various neurodegenerative disorders that affect neurotransmitter secretion. Therefore, the research is timely and has a great significance.
- (b) We appreciate the positive response.

Comment 2

- (a) Nonetheless, it is not clear the novelty of the work. Single cell amperometry and the use of carbon fibre electrodes, has been already widely reported. As such, the novelty lies in the development of the device and the method for rapid data interpretation. While these features are important in advancing research in the field, the level of innovation may not at the level of publication in a journal like Nature Communications. It is also not clear what is the key novelty compared to other CMOS-based bioelectronic systems for single-cell amperometry already reported in the literature.
- (b) We agree with the reviewer's comment that the key novelty was not clearly described. We have made additions to the manuscript in the introduction to better highlight the key features of this work compared to previous studies. In short, the key novelty of this work is the unprecedented noise level of such a high-density system which is achieved by the new post-CMOS approach as well as the high-quality electronics. Also, we report the first detection of PC-12 quantal secretion at single-vesicle resolution using the integrated CMOS sensor.
- (c) The following is the new addition to the manuscript:

On page 4:

The presented post-CMOS processing has been modified from a previous work¹⁸ to improve the noise performance of each electrochemical sensor through the use of a silicon dioxide insulation layer that defines the effective area of the electrodes throughout the array, thus permitting the developed system to record the small amperometric signals from single vesicle secretions. With the low-noise amplifiers in the CMOS chip and the improved noise performance due to the new post-CMOS process, this work presents the first single-vesicle recordings from pheochromocytoma (PC-12) cells using monolithically integrated CMOS biosensors. The presented silicon-based bioelectronics system can capture a large pool of quantal events in the span of minutes, providing rapid results compared to the conventional amperometry technique that can take months to amass a comparable dataset.

Comment 3

- (a) The authors admit that their device is not capable of true single-cell monitoring (Fig. 2c) and report the issue of variability of electrode coverage by the settled cells. While they proposed several approaches to overcome these issues, they should provide additional evidence by testing at least some of these strategies to demonstrate their effectiveness and stress the relevance of the work presented. As it stands the work looks incomplete.
- (b) We agree that the current writing in the discussion gives an impression of incompleteness. However, that is not the case; our intention was to provide alternative options for different applications. Different surface structures can be ideal depending on the application. In our previous publication, we have demonstrated one of the strategies (single-cell SU-8 traps) to perform the true single-cell monitoring, which can be easily adapted into this system as well. In the presented work, the main

focus is to demonstrate one of the best noise performances to achieve successful recordings of neurotransmitter release from PC-12 cells. PC-12 has relatively smaller quantal size compared to chromaffin cells and therefore their quantal release is more challenging to measure.

Despite the difficulty associated in working with PC-12, the cell line is widely used because it is significantly easier to culture and manipulate compared to other primary cells. Unlike chromaffin cells, PC-12 is capable of generating neural network on the glass surface when given proper conditions. Any obstructive structure, such as SU-8 traps, can prevent their natural tendency to form neural networks between cells. Therefore, in certain applications including PC-12 cells and neurons, measurements from truly-isolated cells are not ideal. The presented work offers single-cell monitoring capability in combination with microscopic imaging for those cells which must be cultured on a flat surface.

(c) The following section is added to the Discussion section.

On page 10:

→Removed paragraph: “Despite the agreement of the effects of L-Dopa treatment to previous studies, , the device presented in this paper is not capable ... epoxy-based photoresist, SU-8, for both bovine chromaffin cells^{26,38,39}a and human neuroblastoma cells, SH-SY5Y²².”

→New paragraph: **The presented data supports the application of the presented system for studies that require single-vesicle resolution or high-throughput, and high-temporal resolution, recordings of any electroactive molecules that undergo a redox reaction below the potential that causes electrolysis of water (1.23V). The presented work, in combination with microscopic imaging, offers single-cell monitoring capability. The flatness of the chip’s surface offers an excellent environment for neuronal cells, or model cells such as PC-12, to create neural networks. Different surface structures may be ideal depending on the application and truly-isolated single-cell recordings can be accomplished with minor modifications to the design, such as using a thick insulation layer to only permit single cells to each electrode. We have demonstrated this technique in our previous work using SU-8¹⁸. However, by incorporating this insulation layer onto a device, the array is not ideal for certain applications including the study of neural networks; the large topology of the insulation may introduce challenges when attempting to culture a neural network on the surface of the array.**

Comment 4

- (a) Overall the manuscript is well written, but the presentation could be improved by for example removing several repetitions along the manuscript that affect the flow. For example, it is mentioned several times that L-Dopa is a popular Parkinson’s disease therapy. Moreover, the flow is broken by several short sentences not clearly connected to each other.
- (b) We agree with the reviewer’s comment. In response, we removed all duplicating information regarding L-Dopa as Parkinson’s disease therapy as well as other sections with repeating data or information.
- (c) Introduction section has been extensively edited in response to Reviewer #1 – Comment 3 and Reviewer #3 – Comment 5. In addition to the changes in the Introduction section, the following is the list of new changes:

On page 8:

→Removed words: “....., a popular Parkinson’s disease therapy,”

On page 10:

→Removed sentences: “Using traditional techniques with CFEs, the effects of L-Dopa treatment on bovine chromaffin cells³³, dopaminergic mouse cells (MN9D)¹, and PC-12 cells^{27,31,34–37}, have been previously studied.”

“The reported L-Dopa modulation of neurotransmitter secretion agrees with the modulation that is observed in our experiments.”

→New sentence: **Using the integrated silicon device, we were able to characterize the effects of L-Dopa treatment at the single vesicle level, and our results agreed with previous studies^{9,23,29–32}.**

On page 12:

→Removed paragraph: “Dysfunctional synaptic transmission is a major determinant of neurodegenerative disorders ... accelerating the studies of various neurodegenerative disorders and their treatments.”

Response to Reviewer #3

Comment 1

- (a) In this manuscript the authors present on-chip measurements of neurotransmitter release using an amperometry CMOS chip. High throughput measurements from PC12 cells are presented showing the expected increase of quantal size after L=Dopa treatment. The described technology is based on some previous work using chromaffin cells and SH-SY5Y cells. The method is very elegant and efficient, and the recordings are of very high quality.
- (b) We appreciate the positive response.

Comment 2

- (a) The results are convincing but unfortunately, the authors do not point out clearly where the novelty lies. The central claim is that the technology with a 1024 on-chip electrodes performs amperometric recording with 10 kHz sampling rate and that the effect of L-Dopa treatment could be demonstrated in a 30 minute recording with very high statistical significance ($p < 0.001$ according to Fig. 4). What aspects are advanced compared to the previous work?
- (b) We agree with the reviewer’s comment that the key novelty was not clearly described, in particular, comparing it to our previous work. We have made additions to the manuscript in the introduction to better highlight the key features of this work compared to previous studies. In short, the key novelty of this work is the unprecedented noise level of such a high-density system which is achieved by the new post-CMOS approach as well as the high-quality electronics. Also, we report the first detection of PC-12 quantal secretion at single-vesicle resolution using the integrated CMOS sensor.
- (c) The following is the new addition to the manuscript.

On page 4:

The presented post-CMOS processing has been modified from a previous work¹⁸ to improve the noise performance of each electrochemical sensor through the use of a silicon dioxide insulation layer that defines the effective area of the electrodes throughout the array, thus permitting the developed system to record the small amperometric signals from single vesicle secretions. With the low-noise amplifiers in the CMOS chip and the improved noise performance due to the new post-CMOS process, this work presents the first single-vesicle recordings from pheochromocytoma (PC-12) cells using monolithically integrated CMOS biosensors. The presented silicon-based bioelectronics system can capture a large pool of quantal events in the span of minutes, providing rapid results compared to the conventional amperometry technique that can take months to amass a comparable dataset.

Comment 3

- (a) The authors should describe more specifically where the specific advance lies. Is the CMOS chip the same as used in their previous work with SH-SY5Y cells (White et al 2018) or are there any modifications? Has the density of electrodes been increased? Is the 5 μm electrode diameter changed from the previous work? What is the background noise? Is it lower than with carbon fibers or previous amperometry chip versions?
- (b) We agree with the reviewer's comment that the advancement of our technology from the previous work is not described in detail. Therefore, we have added new text to highlight the key advancements. We also added a new paragraph dedicated to the comparison of the noise performance to our previous work as well as the recent state-of-the-art sensor.
- (c) The following list of changes is made.

On page 4:

The presented post-CMOS processing has been modified from a previous work¹⁸ to improve the noise performance of each electrochemical sensor through the use of a silicon dioxide insulation layer that defines the effective area of the electrodes throughout the array, thus permitting the developed system to record the small amperometric signals from single vesicle secretions.

On page 7:

The amplifiers throughout the array have an average noise level of $\sim 0.42 \text{ pA}_{\text{RMS}}$ when the device is operating without electrolytic solution at a 10 kHz sampling rate and a bandwidth of $\sim 4.4 \text{ kHz}$ ¹⁸. In comparison, a recent CMOS device has been developed for high-throughput recording of electrochemical current with an array of amplifiers that exhibit a noise level of $1.1 \text{ pA}_{\text{RMS}}$ at a sampling rate of $\sim 10 \text{ kHz}$ with a bandwidth of $\sim 2 \text{ kHz}$ ²⁷. An electrode that is in contact with electrolytic solution suffers from an elevated noise level due to the introduced double-layer capacitance. Our previous design had an electrode area of $15 \mu\text{m} \times 15 \mu\text{m}$, and when electrolytic solution was placed onto the array for electrochemical experiments the noise level was $6.16 \text{ pA}_{\text{RMS}}$ ¹⁸, which is higher than the average amplitude of PC-12 quantal events. The new post-CMOS processing presented in this paper creates an electrode with a diameter of $\sim 5 \mu\text{m}$ and produces an average noise of $\sim 0.9 \text{ pA}_{\text{RMS}}$ under the same conditions. In comparison, a study using a $5 \mu\text{m}$ diameter CFE with the Axopatch 200B amplifier, a high-quality electrophysiology amplifier, had a noise level of $\sim 1.4 \text{ pA}_{\text{RMS}}$ with a 10 kHz bandwidth²⁴. Our electrochemical sensor array exhibits a comparable noise level to a high-quality electrophysiology setup, therefore enabling high-throughput recordings of quantal events from dopaminergic cells, such as PC-12 cells.

Comment 4

- (a) A potential novelty is that the measurements were performed using PC12 cells, which have much smaller quantal size than chromaffin cells. Is this the first report demonstrating on-chip amperometry recordings using PC12 cells? Is there a new record to detect small quantal size (the authors state 187 molecules). What is the "resolution", i.e. what is the error bar for individual quantal size determination ($187 \pm ???$ Molecules). It should be described clearly what the technology can achieve beyond high throughput.
- (b) We are the first to measure PC-12 neurotransmitter secretion using an integrated CMOS sensor. Also, this work is the first to report high-throughput recordings of PC-12's quantal secretion (demonstrated as many as 63 active cells simultaneously; capable of 1024 parallel recordings). The noise performance of the new device is comparable to that of CFE and high-quality patch-clamp amplifiers; however, we are not attempting to report record level detections.

The quantal analysis is done by a software (Quantal Analysis) developed by Dr. David Sulzer which is the main method agreed upon within the field of single-cell amperometry. The typical approach is to use a set of criteria (such as amplitude threshold or shape) to automatically detect the amperometric spikes for data analysis and during the process, it also collects very small spikes with small quantal size. The unique shape from the quantal events do support that these are real events, but it is difficult to conclude what is the real smallest quantal events detected. It is important to note that the analysis is always done with a large pool of events to draw statistical significances and the analysis criterion is set consistently to prevent any bias.

We typically do not make remarks on what the detection limit is in relations to quantal size, and we have mistakenly made claims about 187 molecules. We will eliminate the text related to the smallest quantal size measured during this work. Instead, we can estimate the detection limit based on the noise performance of the device as well as the typical half-width of PC-12 quantal secretions.

- (c) New text has been added to empathize the novelty of the presented work, as a part of the response for Reviewer #3 – Comment 2. Also, we added new test to estimate the detection limit on quantal events based on the noise performance and the typical half-width of PC-12 secretions.

On page 7:

→New sentences: **Based on the noise level of the presented device (~0.9 pA_{RMS}) and the typical half-width of PC-12 cells (~5 ms), we can estimate the smallest quantal size that can be detected using this system. Because the integrated noise level for 5-ms events is about 0.14 pA_{RMS}, the quantal size equivalent noise level for 5-ms signal is ~2000 molecules.**

Comment 5

- (a) The introduction is too long describing many results obtained with conventional amperometry, which is not highly relevant for this manuscript. For instance, most of what is on page 3 could be deleted.
- (b) We agree with the reviewer's comment. We have made extensive revision to the introduction section to remove duplicating information as well as loosely-relevant texts.
- (c) Detailed changes in the Introduction are listed in the response to Reviewer #1 – Comment 3. In addition, deletions in Page 3 in response to this comment are listed below:

On page 2:

→Removed sentences: “Using traditional amperometry techniques researchers have discovered that extracellular calcium ... are secreted during exocytosis events and a lack of the C-terminus leads to prolonged secretion events¹².”

“Amperometry is also used to study the effects of various treatments that modulate exocytosis characteristics, ... reduces the quantal size of the exocytotic events while curcumin accelerates the exocytotic process¹⁷.”

“To monitor the concentration of neurotransmitters, such as acetylcholine, in vivo to enhance therapies for neurological diseases, surface modified electrodes are being developed to enable amperometry monitoring of biomolecules that are not inherently electroactive¹⁹.”

Comment 6

- (a) The paragraph preceding subheading “Identifying effect of Parkinson's disease therapy to quantal release in minutes” duplicates some of what is in the following section. It may be more logical to remove in that paragraph all text related to L-Dopa and focus on untreated cell recordings and issues of sensitivity and resolution of amperometric spike parameters.

- (b) We agree with the comment. In response, we have deleted text in “On-chip recordings at single-vesicle resolution” section that pertained to L-Dopa treatments and the new text focuses on untreated cell recordings. All the discussion related to L-Dopa is now in “Rapid characterization ...” section.
- (c) The following change have been made.

On page 8:

→Removed sentences: “In the 14-minute recording for the L-Dopa treated cells, 3534 quantal events are detected (Fig. 4). The ... quantal size for the treated group is larger as expected compared to the control group for both extremes.”

REVIEWERS' COMMENTS

Reviewer #2 (Remarks to the Author):

The authors have addressed all the points that the reviewers raised. Nonetheless, I still have the following comments :

The authors should further improve the use of the language throughout the manuscript. For example: 1) No contracted form should be used (see for example the use of 'isn't' in the abstract); 2) The edited sentence 'Chemical transmissions at synapses are an essential mechanism in neuronal networks, because it facilitates the propagation of information throughout the nervous system for body functions, memories, and emotions' should be corrected as: 'Chemical transmissions at synapses are an essential mechanism in neuronal networks, because they facilitate the propagation of information throughout the nervous system for body functions, memories, and emotions.' Overall a careful reading of the manuscript is suggested.

Consider rephrasing the following sentence :

'To measure the secretion of electroactive neurotransmitters, such as dopamine, serotonin, epinephrine, and norepinephrine, the amperometric setup consists of a reference electrode and a working electrode.'

The following sentence should be broken up in at least two:

'Amperometric recordings using carbon fiber electrodes (CFEs) have been pivotal in studying the fundamental mechanisms of vesicle fusion^{2,4-7}, as well as the molecular effects of neurological diseases and pharmacological treatments that modulate the exocytosis process⁸⁻¹⁰, over many decades.'

The following sentence is very bold and unless appropriately referenced is worthless.

'Despite the significance of the studies conducted and the discoveries made using amperometry, the technique is extremely labor-intensive and can take months to years to research the effects of neurological diseases or pharmacological treatments'

On page 10: 'The presented data supports' please note that data is a plural.

The following sentence should be properly referenced: 'The presented silicon-based bioelectronics system can capture a large pool of quantal events in the span of minutes, providing rapid results compared to the conventional amperometry technique that can take months to amass a comparable dataset.'

The authors state: ' Different surface structures may be ideal depending on the application and truly-isolated single-cell recordings can be accomplished with minor modifications to the design, such as using a thick insulation layer to only permit single cells to each electrode. We have demonstrated this technique in our previous work using SU-818. However, by incorporating this insulation layer onto a device, the array is not ideal for certain applications including the study of neural networks; the large topology of the insulation may introduce challenges when attempting to culture a neural network on the surface of the array.'

What do the authors suggest then as a suitable strategy to enhance the surface strategy? While the authors have strategically rephrased their previous sentence, I still feel that this study is uncomplete. This is fine provided that the work has sufficient novelty to justify the urgency for its publication in its state.

Nonetheless, I am still not convinced on the actual novelty of this work, despite the fact that the authors have clarified this point better than in the previous submission.

Reviewer #3 (Remarks to the Author):

The authors have done an excellent job with the revisions. My concerns have overall been very well addressed. The authors now provide a clear and quantitative description and discussion of the advances achieved with the device and the experiments described in this manuscript. However, the phrase "the typical half-width of PC12 cells" should be replaced by "the typical half-width of amperometric spikes recorded from PC12 cells".

Response to referees

Response to Reviewer #2

Comment 1

- (a) The authors have addressed all the points that the reviewers raised.
- (b) We appreciate the positive response.

Comment 2

- (a) The authors should further improve the use of the language throughout the manuscript. For example:
 - 1) No contracted form should be used (see for example the use of 'isn't' in the abstract);
 - 2) The edited sentence 'Chemical transmissions at synapses are an essential mechanism in neuronal networks, because it facilitates the propagation of information throughout the nervous system for body functions, memories, and emotions' should be corrected as: 'Chemical transmissions at synapses are an essential mechanism in neuronal networks, because they facilitate the propagation of information throughout the nervous system for body functions, memories, and emotions.' Overall a careful reading of the manuscript is suggested.
 - 3) Consider rephrasing the following sentence: 'To measure the secretion of electroactive neurotransmitters, such as dopamine, serotonin, epinephrine, and norepinephrine, the amperometric setup consists of a reference electrode and a working electrode.'
 - 4) The following sentence should be broken up in at least two: 'Amperometric recordings using carbon fiber electrodes (CFEs) have been pivotal in studying the fundamental mechanisms of vesicle fusion^{2,4-7}, as well as the molecular effects of neurological diseases and pharmacological treatments that modulate the exocytosis process⁸⁻¹⁰, over many decades.'
 - 5) On page 10: 'The presented data supports' please note that data is a plural.
- (b) We agree with the comment. In response, we have made revisions to the writings. We went through the entire document to remove contracted forms. We also address the specific sentences the reviewer highlighted for change. Also, we have made additional effort to improve language throughout the paper.
- (c) The list of changes is below:

On page 1:

Traditional single-cell amperometry is a powerful technique for studying the molecular mechanisms of exocytosis, but **it is** both costly and labor-intensive to accumulate statistically significant data.

On page 2:

Chemical transmissions at synapses are an essential mechanism in neuronal networks, because they facilitate the propagation of information throughout the nervous system for body functions, memories, and emotions.

On page 2:

In the amperometric setup, an electric potential is held between two electrodes, a reference electrode and a working electrode. This setup allows electroactive molecules, such as dopamine, serotonin, epinephrine, and norepinephrine, to undergo either oxidation, the release of electrons from the molecule, or reduction, the acquisition of electrons to the molecule, and generates a current that is measurable.

On page 2:

Amperometric recordings using carbon fiber electrodes (CFEs) have been pivotal in studying the fundamental mechanisms of vesicle fusion^{2,4-7} over many decades. Amperometry using CFEs have also been crucial for studying the molecular effects of neurological diseases and pharmacological treatments that modulate the exocytosis process⁸⁻¹⁰.

On page 3:

The high sensitivity and temporal resolution of amperometry has enabled *in vivo* animal study of monitoring the concentration of nitric oxide, which is linked to the neuronal loss¹⁵.

On page 6:

CMOS fabrication foundries typically use an aluminum-copper (Al/Cu) alloy for the top metal layer, which is inadequate for neurotransmitter detection as it is highly reactive to electrolytic solutions.

On page 6:

This highly reactive metal alloy results in significant current offsets, thus increasing the shot noise of the amplifiers.

On page 6:

The chip is first processed using a negative photoresist, NR9-1500PY, to create a sacrificial layer that defines where metal deposition is undesirable.

On page 8:

Quantal events from the attached cells are recorded in parallel (Fig. 3a) with the electrodes throughout the array held at a potential of ~800 mV with respect to the reference electrode and the amplifiers operating at a 10 kHz sampling rate.

On page 10:

Using the presented CMOS integrated electrochemical sensors, we are able to perform hundreds of amperometric recordings in minutes as opposed to weeks using the conventional CFE approach²⁸.

On page 10:

The presented data, specifically the validated changes in I_{\max} , $t_{1/2}$, and Q quantal characteristics after L-Dopa treatment, supports the application of the presented system for studies that require single-vesicle resolution or high-throughput, and high-temporal resolution, recordings of any electroactive molecules that undergo a redox reaction below the potential that causes electrolysis of water (1.23V).

- (a) The following sentence is very bold and unless appropriately referenced is worthless. ‘Despite the significance of the studies conducted and the discoveries made using amperometry, the technique is extremely labor-intensive and can take months to years to research the effects of neurological diseases or pharmacological treatments’
The following sentence should be properly referenced: ‘The presented silicon-based bioelectronics system can capture a large pool of quantal events in the span of minutes, providing rapid results compared to the conventional amperometry technique that can take months to amass a comparable dataset.’
- (b) We agree with the reviewer’s comment. The reference for this sentence was missing in the original submission. In response, we added the references to support the statement. We’ve also made a bold statement that the CFE studies can take from months to years, but we have adjusted the timeline according to the references (weeks to months).
- (c) The following change has been made to the manuscript.

On page 3:

Despite the significance of the studies conducted and the discoveries made using amperometry, the CFE technique is labor-intensive¹⁸⁻²⁴, time consuming^{21,25-27}, and can take from weeks to several months²⁸ to research the effects of neurological diseases or pharmacological treatments.

On page 4:

The presented silicon-based bioelectronics system can capture a large pool of quantal events in the span of minutes, providing rapid results compared to the conventional amperometry technique that can take from weeks to months to amass a comparable dataset²⁸.

Comment 4

- (a) The authors state: ‘Different surface structures may be ideal depending on the application and truly-isolated single-cell recordings can be accomplished with minor modifications to the design, such as using a thick insulation layer to only permit single cells to each electrode. We have demonstrated this technique in our previous work using SU-8¹⁸. However, by incorporating this insulation layer onto a device, the array is not ideal for certain applications including the study of neural networks; the large topology of the insulation may introduce challenges when attempting to culture a neural network on the surface of the array.’
What do the authors suggest then as a suitable strategy to enhance the surface strategy? While the authors have strategically rephrased their previous sentence, I still feel that this study is incomplete. This is fine provided that the work has sufficient novelty to justify the urgency for its publication in its state. Nonetheless, I am still not convinced on the actual novelty of this work, despite the fact that the authors have clarified this point better than in the previous submission.
- (b) For the PC-12 application on CMOS, the only suitable approach is to restrict the size of electrodes to reduce the noise level such that it closely matches that of a high-quality CFE electrophysiology setup. Notably, PC-12 exhibits smaller amperometric spikes compared to other commonly used cell models, such as bovine chromaffin cells. In this project, we successfully measured PC-12 amperometric spikes by using SiO₂ insulation, allowing us to confine the electrode size to ~ 5 μm which is nearly identical to CFE microelectrodes. Our result showed that we can solve even single-vesicle secretion events.
- (c) The following change has been made to the manuscript.

On page 12:

The presented work enables single-vesicle level amperometric recordings from PC-12 on the CMOS-based biosensor by restricting the size of electrodes, thus reducing the noise level such that it closely matches that of the high-quality CFE electrophysiology setup. This work presents a high-throughput analysis platform for electrochemical detection of quantal secretion with single-vesicle resolution.

Response to Reviewer #3

Comment 1

- (a) The authors have done an excellent job with the revisions. My concerns have overall been very well addressed. The authors now provide a clear and quantitative description and discussion of the advances achieved with the device and the experiments described in this manuscript.
- (b) We appreciate the positive response.

Comment 2

- (a) However, the phrase “the typical half-width of PC12 cells” should be replaced by “the typical half-width of amperometric spikes recorded from PC12 cells”.
- (b) We agree with the reviewer’s comment and appreciate the given suggestion. We made the recommended revision.
- (c) The following change has been made to the manuscript.

On page 7:

Based on the noise level of the presented device (~ 0.9 pA_{RMS}) and the typical half-width of **amperometric spikes recorded from** PC-12 cells (~ 5 ms), we can estimate the smallest quantal size that can be detected using this system.